REGISTERED REPORT

# Registered report: Widespread potential for growth factor-driven resistance to anticancer kinase inhibitors

Edward Greenfield[1], Erin Griner[2], Reproducibility Project: Cancer Biology*[†]

[1]Monoclonal Antibody Core Facility, Dana-Farber Cancer Institute, Boston, United States; [2]University of Virginia, Charlottesville, United States

## REPRODUCIBILITY PROJECT CANCER BIOLOGY

**Abstract** The Reproducibility Project: Cancer Biology seeks to address growing concerns about reproducibility in scientific research by conducting replications of 50 papers in the field of cancer biology published between 2010 and 2012. This Registered Report describes the proposed replication plan of key experiments from 'Widespread potential for growth-factor-driven resistance to anticancer kinase inhibitors' by Wilson and colleagues, published in Nature in 2012 (*Wilson et al., 2012*). The experiments that will be replicated are those reported in Figure 2B and C. In these experiments, Wilson and colleagues show that sensitivity to receptor tyrosine kinase (RTK) inhibitors can be bypassed by various ligands through reactivation of downstream signaling pathways (Figure 2A; *Wilson et al., 2012*), and that blocking the receptors for these bypassing ligands abrogates their ability to block sensitivity to the original RTK inhibitor (Figure 2C; *Wilson et al., 2012*). The Reproducibility Project: Cancer Biology is a collaboration between the Center for Open Science and Science Exchange, and the results of the replications will be published by *eLife*.

**\*For correspondence:** fraser@ scienceexchange.com

**Group author details**
[†]Reproducibility Project: Cancer Biology
See page 16

**Reviewing editor**: Joan Massagué, Memorial Sloan-Kettering Cancer Center, United States

## Introduction

A recurring theme in treatment of cancer is the acquisition of drug resistance. The effectiveness of therapies targeting specific mutations in receptor tyrosine kinases (RTKs) is limited by the acquisition of resistance to the drugs over the course of treatment (*Mok et al., 2009*; *Camidge et al., 2014*). Resistance can be acquired through new mutations that block the action of the RTK inhibitors or their uptake and/or genetic amplification of downstream target genes of the RTK (*Chen and Fu, 2011*; *Garrett and Arteaga, 2011*; *Sequist et al., 2011*; *Gainor and Shaw, 2013*; *Yang, 2013*). Several studies, including this work by Wilson and colleagues, elucidated another mechanism for this acquisition of resistance: the engagement of parallel RTK signaling pathways that converge on common downstream survival signals via signals from the tumor microenvironment. In this study, Wilson and colleagues examined several cancer cell lines for ligand-mediated drug resistance (*Wilson et al., 2012*).

In Figure 2B/C, Wilson and colleagues demonstrated that resistance to primary kinase inhibitor treatment can be induced by the addition of rescuing ligands that activate the PI(3)K–AKT and MAPK pro-survival signaling pathways. This resistance can be overcome with the addition of an appropriate secondary kinase inhibitor. Three different cancer cell line models were used to demonstrate this phenomenon. Treatment of A204 (a *PDGFR* amplified rhabdomyosarcoma cell line) with the ligand FGF activated pFRS2 and pERK, inducing resistance to sunitinib. The addition of a secondary kinase inhibitor, PD173074, blocked FGF-induced pFRS2 and pERK activation, restoring sensitivity to sunitinib. The treatment of M14 (a *BRAF*-mutated melanoma cell line) with the ligand NRG1 activated pHER3 and pAKT, inducing partial resistance to PLX4032. The addition of a secondary kinase inhibitor, lapatinib, blocked NRG1-induced pHER3 and pAKT activation, restoring sensitivity to PLX4032. Treatment of KHM-3S (an *EGFR*-mutated small cell lung cancer cell line) with the ligand HGF

activated pMET and pERK, inducing resistance to Erlotinib. The addition of a secondary kinase inhibitor, crizotinib, blocked HGF-induced pMET and pERK activation, restoring sensitivity to erlotinib.

The cell viability assays examining drug sensitivity and the Western blots examining levels of phosphorylated kinases in Figures 2B and 2C, respectively, are the key experiments that demonstrate that growth factor ligands can reactivate downstream signaling components important for cancer cell survival, causing resistance to anticancer kinase inhibitors (*Wilson et al., 2012*). These experiments are replicated in Protocols 1 and 2.

Two studies published around the same time as the work of Wilson and colleagues also support the proposed mechanism of acquired resistance to RTK inhibition by signaling from the tumor microenvironment. Straussman and colleagues demonstrated that HGF signaling derived from the tumor microenvironment could bypass EGFR inhibition by activation of MET signaling (*Straussman et al., 2012*, also included for replication in the Reproducibility Project: Cancer Biology), and Harbinski and colleagues, in an approach similar to Wilson and colleagues, showed that multiple growth factor ligands could 'bypass' inhibitor-targeted RTKs (*Harbinski et al., 2012*).

Since the publication of Wilson and colleagues' work, several publications have reported similar results to those being replicated in Protocols 1 and 2. Similar to the experiments with A204 cells above, Welti and colleagues demonstrated that FGF ligands could induce resistance to sunitinib, which could be reversed by the addition of PD173074 (*Welti et al., 2011*). These experiments were performed in HUVEC cells, whereas A204 cells were used in the study being replicated. Similar to the experiments on M14 cells above, Montero-Conde and colleagues showed that NRG1 ligand could activate pHER3 and pAKT in the presence of PLX4032, and this activation could be reversed by the addition of lapatinib (*Montero-Conde et al., 2013*). These experiments were performed in 8505C cells, whereas M14 cells were used in the study being replicated. Similar to the experiments performed on KHM-S3 cells above, several groups have demonstrated that HGF ligand can induce resistance to erlotinib and that this resistance can be reversed by the addition of crizotinib (*Nakagawa et al., 2012*; *Nakade et al., 2014*). These experiments were performed in PC-9 and HCC827 cells, whereas KHM-3S cells were used in the study being replicated.

## Materials and methods

Unless otherwise noted, all protocol information was derived from the original paper, references from the original paper, or information obtained directly from the authors. An asterisk (*) indicates data or information provided by the Reproducibility Project: Cancer Biology core team. A hashtag (#) indicates information provided by the replicating lab.

### Protocol 1: Cell viability assays

This protocol describes cell viability assays to determine the $IC_{50}$ values of three cancer cell lines treated with primary kinase inhibitor alone, primary kinase inhibitor in combination with rescuing ligand, and primary kinase inhibitor in triple combination with rescuing ligand and a drug targeting the rescuing ligand's receptor tyrosine kinase (RTK) (termed the secondary kinase inhibitor) (Figure 2B).

### Sampling

- The original data presented is qualitative, and the authors were unable to share the raw data values with the RP:CB core team. This prevents power calculations being performed a priori to determine the sample size (number of biological replicates). In order to determine an appropriate number of replicates to perform initially, we have estimated the sample sizes required based on a range of potential variance. We will also determine the sample size *post hoc* as described in Power Calculations.

  1. Please see Power Calculations for details.

- Each experiment has three cohorts. In each cohort, a dilution series of the primary kinase inhibitor ($10^{-4}$, $10^{-3}$, $10^{-2}$, $10^{-1}$, $10^{0}$, and $10^{1}$ µM) is run three times; once alone, once with the rescuing ligand, and once with both the rescuing ligand and the secondary kinase inhibitor. The effect of the secondary kinase inhibitor alone will also be assessed. Each condition will be run in triplicate.

  1. Cohort 1: A204 cell line.

     - Media only [additional].
     - Vehicle control.

- 0.001 µM–10 µM sunitinib + no ligand.
- 0.001 µM–10 µM sunitinib + 50 ng/ml FGF.
- 0.001 µM–10 µM sunitinib + 50 ng/ml FGF + 0.5 µM PD173074.
- 0.5 µM PD173074 + no ligand [additional].

2. Cohort 2: M14 cell line.

- Media only [additional].
- Vehicle control.
- 0.001 µM–10 µM PLX4032 + no ligand.
- 0.001 µM–10 µM PLX4032 + 50 ng/ml NRG1.
- 0.001 µM–10 µM PLX4032 + 50 ng/ml NRG1 + 0.5 µM lapatinib.
- 0.5 µM lapatinib + no ligand [additional].

3. Cohort 3: KHM-3S cell line.

- Media only [additional].
- Vehicle control.
- 0.001 µM–10 µM erlotinib + no ligand.
- 0.001 µM–10 µM erlotinib + 50 ng/ml HGF.
- 0.001 µM–10 µM erlotinib + 50 ng/ml HGF + 0.5 µM crizotinib.
- 0.5 µM crizotinib + no ligand [additional].

## Materials and reagents

| Reagent | Type | Manufacturer | Catalog # | Comments |
|---|---|---|---|---|
| 96-well tissue culture plates | Materials | Corning (Sigma-Aldrich) | CLS3516 | Original unspecified |
| KHM-3S cells | Cells | JCRB Cell Bank | JCRB0138 | Original source of the cells unspecified |
| A204 | Cells | ATCC | HTB-82 | Original source of the cells unspecified |
| M14 | Cells | ATCC | HTB-129* | Original source of the cells unspecified |
| Lapatinib | Drug | LC Laboratories | L-4804 | Original formulation unspecified |
| Crizotinib | Drug | Sigma-Aldrich | PZ0191 | Originally from Selleck Chemicals |
| PD173074 | Drug | Sigma-Aldrich | P2499 | Originally from Tocris Bioscience |
| PLX4032 | Drug | Active Biochem | A-1130 | |
| Sunitinib | Drug | Sigma-Aldrich | PZ0012 | Originally from Selleck Chemicals, formulation unspecified |
| Erlotinib | Drug | LC Laboratories | E-4007 | |
| HGF | Ligand | Sigma-Aldrich | H5791 | Originally obtained from Peprotech |
| FGF-basic | Ligand | Sigma-Aldrich | F0291 | Originally obtained from Peprotech |
| NRG1-β1 | Ligand | Novus Biologicals | H00003084-P01 | Originally obtained from R&D Systems |
| RPMI 1640 | Media | Sigma-Aldrich | R8758 | Originally from Gibco, formulation unspecified |
| FBS | Reagent | Sigma-Aldrich | F4135 | Originally from Gibco |

*Table 1. Continued on next page*

*Table 1. Continued*

| Reagent | Type | Manufacturer | Catalog # | Comments |
|---|---|---|---|---|
| Penicillin | Antibiotic | Sigma-Aldrich | P4458 | Original unspecified |
| Streptomycin | Antifungal | | | Original unspecified |
| Paraformaldehyde | Reagent | Sigma-Aldrich | 158127 | Original unspecified |
| Syto 60 | Reagent | Life Technologies | S11342 | Original unspecified |
| Odyssey scanner | Equipment | LiCOR | | |
| Odyssey application software | Software | LiCOR | | |

*The breast cancer cell line MDA-MB-435 has been shown to be mislabeled; it is in fact identical to the M14 melanoma cell line (**Rae et al., 2007**; **Chambers, 2009**; **Holliday and Speirs, 2011**).

## Procedure

### Notes

- All cells will be sent for mycoplasma testing and STR profiling.
- Medium for all cell lines: RPMI 1640 supplemented with 10% FBS, 50 U/ml penicillin, and 50 µg/ml streptomycin.
- Cells maintained at 37°C in a humidified atmosphere at 5% $CO_2$.

1. Seed 3000–5000 cells per well into 96-well plates. For each condition replicate seed 1 well as the media control, 1 well as the vehicle control, 1 well for treatment with the secondary kinase inhibitor alone, and 6 wells per concentration curve ($10^{-4}$, $10^{-3}$, $10^{-2}$, $10^{-1}$, $10^{0}$, and $10^{1}$ µM), of which there are three.

    a. 6 wells per concentration curve × 3 concentration curves = 18 wells + 3 wells = 21 wells per cohort.

2. 18–24 hr after seeding treat 3 wells per condition with appropriate treatment (see Sampling).

    a. Lab will record the vehicle used to solubilize the drugs.

3. 72 hr after treatment, fix cells in 4% paraformaldehyde (PFA).

    a. Lab will record the PFA incubation time.

4. Stain with Syto 60 according to the manufacturer's recommendations and assay cell number using an Odyssey with Odyssey Application Software.

    a. Include empty wells and media only wells.

5. Calculate cell viability by dividing the fluorescence from the drug-treated cells by the fluorescence from the control (vehicle) treated cells. Fit normalized data to a sigmoidal dose–response curve.

    a. Also calculate the effect of vehicle by dividing the fluorescence from the control vehicle cells by the fluorescence from the media only treated cells [additional control].
    b. Determine the $IC_{50}$ values for each curve.
    c. Lab will document the software used to fit the data to a sigmoidal dose–response curve and calculate the $IC_{50}$ values.

6. Repeat independently two additional times.

### Deliverables

- Data to be collected:

1. Raw fluorescence data and calculated cell viability.
2. Semi-logarithmic graph for each condition of primary kinase inhibitor (log) vs normalized cell viability (linear) for each cell line [comparable to Figure 2B].
3. Calculated $IC_{50}$ for each condition.

## Confirmatory analysis plan

- Statistical analysis of the Replication Data:

    1. For each cell line compare the $IC_{50}$ of primary kinase inhibitor alone, primary kinase inhibitor + ligand, and primary kinase inhibitor + ligand + secondary kinase inhibitor.

        • ANOVA.

- Meta-analysis of original and replication attempt effect sizes:

    1. We will plot the replication data (mean and 95% confidence interval) and will include the original data point, calculated directly from the representative image in Figure 2B, as a single point on the same plot for comparison.

## Known differences from the original study

- We are including two additional control conditions;

    1. Media alone.

        a. To provide a baseline.

    2. Treatment of the cells with the secondary kinase inhibitor alone.

        a. To assess any effects, the secondary kinase inhibitor may be independent of the ligand and primary kinase inhibitor.

## Provisions for quality control

- All data obtained from the experiment—raw data, data analysis, control data, and quality control data—will be made publicly available, either in the published manuscript or as an open access dataset available on the Open Science Framework (https://osf.io/h0pnz/).
- Cell lines will be validated by STR profiling and screened for mycoplasma contamination.
- A lab from the Science Exchange network with extensive experience in conducting cell viability assays will perform these experiments.

## Protocol 2: Western blot assays

This protocol describes Western blot assays to determine the levels of activated phosphorylated signaling pathways in three cancer cell lines treated with primary kinase inhibitor alone, primary kinase inhibitor in combination with rescuing ligand, and primary kinase inhibitor in triple combination with rescuing ligand and a drug targeting the rescuing ligand's receptor tyrosine kinase (RTK) (termed the secondary kinase inhibitor) (Figure 2C).

### Sampling

- The original data presented is qualitative. This prevents power calculations being performed a priori to determine the sample size (number of biological replicates). In order to determine an appropriate number of replicates to perform initially, we have estimated the sample sizes required based on a range of potential variance. We will also determine the sample size *post hoc* as described in Power Calculations.

    1. Please see Power Calculations for details.

- Each experiment has three cohorts. Each cohort will consist of cells treated with media alone, with vehicle alone, with the primary kinase inhibitor, with primary kinase inhibitor and the rescuing ligand and with the primary kinase inhibitor, the rescuing ligand and the secondary kinase inhibitor. The effect of the secondary kinase inhibitor alone will also be assessed. Each condition will be run once (i.e., no technical replicates will be performed).

1. Cohort 1: A204 cell line.

   - Media only [additional].
   - Vehicle control.
   - 1 µM sunitinib + no ligand.
   - 1 µM sunitinib + 50 ng/ml FGF.
   - 1 µM sunitinib + 50 ng/ml FGF + 0.5 µM PD173074.
   - 1 µM PD173074 + no ligand [additional].

2. Cohort 2: M14 cell line.

   - Media only [additional].
   - Vehicle control.
   - 1 µM PLX4032 + no ligand.
   - 1 µM PLX4032 + 50 ng/ml NRG1.
   - 1 µM PLX4032 + 50 ng/ml NRG1 + 0.5 µM lapatinib.
   - 1 µM lapatinib + no ligand [additional].

3. Cohort 3: KHM-3S cell line.

   - Media only [additional].
   - Vehicle control.
   - 1 µM erlotinib + no ligand.
   - 1 µM erlotinib + 50 ng/ml HGF.
   - 1 µM erlotinib + 50 ng/ml HGF + 0.5 µM Crizotinib.
   - 1 µM crizotinib + no ligand [additional].

4. Cohort 4: positive control cell lines.

   - For Cohort 1: HL60 cells treated with FGF [additional control].
   - For Cohort 2: MCF7 cells treated with NRG1 [additional control].
   - For Cohort 3: HEK293 cells treated with HGF [additional control].

   a. Treatment of these cell lines with their cognate growth factor ligands will serve as a positive control for ligand activity.

## Materials and reagents:

| Reagent | Type | Manufacturer | Catalog # | Comments |
|---|---|---|---|---|
| 96-well Tissue culture plates | Materials | Corning (Sigma-Aldrich) | CLS3596 | Original unspecified |
| 6-well tissue culture plates | Materials | Corning (Sigma-Aldrich) | CLS3516 | Original unspecified |
| KHM-3S cells | Cells | JCRB Cell Bank | JCRB0138 | Original source of the cells unspecified |
| A204 cells | Cells | ATCC | HTB-82 | Original source of the cells unspecified |
| M14 cells | Cells | ATCC | HTB-129 | Original source of the cells unspecified |
| HL60 cells | Cells | ATCC | CCL-240 | |
| MCF7 cells | Cells | ATCC | HTB-22 | |
| HEK293 cells | Cells | ATCC | CRL-1573 | |
| Lapatinib | Drug | LC Laboratories | L-4804 | Original formulation unspecified |
| Crizotinib | Drug | Sigma-Aldrich | PZ0191 | Originally from Selleck Chemicals |
| PD173074 | Drug | Sigma-Aldrich | P2499 | Originally from Tocris Bioscience |

*Table 2. Continued on next page*

*Table 2. Continued*

| Reagent | Type | Manufacturer | Catalog # | Comments |
|---|---|---|---|---|
| PLX4032 | Drug | Active Biochem | A-1130 | |
| Sunitinib | Drug | Sigma-Aldrich | PZ0012 | Originally from Selleck Chemicals, formulation unspecified |
| Erlotinib | Drug | LC Laboratories | E-4007 | |
| HGF | Ligand | Sigma-Aldrich | H5791 | Originally obtained from Peprotech |
| FGF-basic | Ligand | Sigma-Aldrich | F0291 | Originally obtained from Peprotech |
| NRG1-β1 | Ligand | Novus Biologicals | P1426 | Originally obtained from R&D Systems |
| RPMI 1640 | Media | Sigma-Aldrich | R8758 | Originally from Gibco, formulation unspecified |
| FBS | Reagent | Sigma-Aldrich | F4135 | Originally from Gibco |
| Penicillin | Antibiotic | Sigma-Aldrich | P4458 | Original unspecified |
| Streptomycin | Antifungal | | | Original unspecified |
| Halt protease and phosphatase cocktail inhibitor | Reagent | Thermo Scientific | 78440 | |
| Image J | Software | National Institutes of Health (NIH) | N/A | |
| p-PDGFRα | Antibody | Santa Cruz | SC-12911 | 190 kDa |
| PDGFRα | Antibody | Cell Signaling | 5241 | 190 kDa |
| p-AKT S473 | Antibody | Invitrogen | 44-621 G | 65 kDa |
| AKT | Antibody | Cell Signaling | 9272 | 65 kDa |
| p-ERK T202/Y204 | Antibody | Cell Signaling | 9101 | 44,42 kDa |
| ERK | Antibody | Cell Signaling | 9102 | 44,42 kDa |
| pFRS2α Y196 | Antibody | Cell Signaling | 3864 | 85 kDa |
| FRS2α | Antibody | Santa Cruz | SC-8318 | 85 kDa |
| β-tubulin | Antibody | Cell Signaling | 2146 | 55 kDa |
| pHER3 Y1289 | Antibody | Cell Signaling | 4791 | 185 kDa |
| HER3 | Antibody | Santa Cruz | SC-285 | 185 kDa |
| p-EGFR Y1068 | Antibody | Abcam | ab5644 | 185 kDa |
| EGFR | Antibody | BD Biosciences | 610017 | 185 kDa |
| p-MET Y1234/5 | Antibody | Cell Signaling | 3126 | 145 kDa |
| MET | Antibody | Santa Cruz | SC-10 | 145 kDa |
| Anti-Mouse IgG-HRP | Antibody | Cell Signaling Technology | 7076P2 | Original unspecified |
| Anti-Rabbit IgG-HRP | Antibody | Cell Signaling Technology | 7074P2 | Original unspecified |
| Anti-Goat IgG-HRP | Antibody | Santa Cruz Biotechnology | sc-2020 | Original unspecified |
| Trypsin-EDTA solution (1X) | Reagent | Sigma-Aldrich | T3924 | Original unspecified |
| Dulbecco's Phosphate Buffered Saline | Reagent | Sigma-Aldrich | D1408 | Original unspecified |
| Mini Protean TGX 4–15% Tris-Glycine gels; 15-well; 15 µl | Reagent | Bio-Rad | 456-1086 | Original unspecified |
| 2X Laemmli sample buffer | Reagent | Sigma-Aldrich | S3401 | Original unspecified |
| ECL DualVue Western Markers (15 to 150 kDa) | Reagent | Sigma-Aldrich | GERPN810 | Original unspecified |

*Table 2. Continued on next page*

*Table 2. Continued*

| Reagent | Type | Manufacturer | Catalog # | Comments |
|---|---|---|---|---|
| Nitrocellulose membrane; 0.45 µm, 20 × 20 cm | Reagent | Bio-Rad | 162-0113 | Original unspecified |
| Ponceau S | Reagent | Sigma-Aldrich | P7170 | Original unspecified |
| Tris Buffered Saline (TBS); 10X solution | Reagent | Sigma-Aldrich | T5912 | Original unspecified |
| Tween 20 | Reagent | Sigma-Aldrich | P1379 | Original unspecified |
| Nonfat-Dried Milk | Reagent | Sigma-Aldrich | M7409 | Original unspecified |
| Super Signal West Pico Substrate | Reagent | Thermo-Fisher (Pierce) | 34087 | |

## Procedure

### Notes

- All cells will be sent for mycoplasma testing and STR profiling.
- Medium for cell lines: RPMI 1640 supplemented with 10% FBS, 50 U/ml penicillin, and 50 µg/ml streptomycin.

    1. MCF7 cells and HEK293 cells are maintained in DMEM + 10% FBS.

- Cells maintained at 37°C in a humidified atmosphere at 5% $CO_2$.

    1. Seed cells in plates.

        a. Two control and four experimental wells (6 wells total) are needed for each cell line in Cohorts 1–3.

            i. Lab will determine and record the number of cells seeded and well size used.

        b. *For Cohort 4 seed cells as needed into wells of a 6-well plate.

    2. 18–24 hr after seeding treat wells in Cohorts 1–3 with conditions as described in the Sampling section.

        a. Lab will determine and record vehicle for preparation of drug solutions.
        b. Harvest protein as in Step 5 after 2 hr of treatment.

    3. Simultaneously treat cells in Cohort 4 as follows:

        a. HL60 cells. Note: This protocol is based on *Krejci et al. (2003)*.

            i. Serum starve HL60 cells for 24 hr prior to protein harvesting.

                - Serum starve = DMEM + 0% FBS.

            ii. Treat cells for 10 min with 100 ng/ml FGF.
            iii. Harvest cell lysates as noted in Step 5.

        b. MCF7 cells. Note: This protocol is based on *Sarup et al. (2008)*.

            i. Serum starve cells for 48 hr prior to protein harvesting.

                - Serum starve = DMEM + 0.1% BSA.

            ii. Treat cells with 1 nmol/l NRG1 for 10 min at 37°C.
            iii. Harvest cell lysates as noted in Step 5.

        c. HEK293 cells. Note: This protocol is based on *Wright et al. (2012)*.

            i. Serum starve HEK293 cells for 24 hr prior to protein harvesting.

                - Serum starve = DMEM + 0% FBS.

            ii. Treat cells with 29 ng/ml HGF for 10 min at 37°C.
            iii. Harvest cell lysates as noted in Step 5.

4. #Preparation of cell lysate:

   a. Note: from here on, the replicating lab will use their in-house Western blot protocol, as recommended by the original authors.
   b. Harvest cells from the tissue culture plate using 1× trypsin–EDTA.
   c. Wash cells with 1× cold PBS and spin at 1200 rpm for 5 min.
   d. Decant the PBS and add lysis buffer to the cell pellet and resuspend well.
   e. Incubate at room temperature for 5 min.
   f. Spin solution at 13,000 rpm for 30 min at 4°C using a benchtop centrifuge.
   g. Collect the lysate/protein sample and store at −20°C or −80°C for later use.

5. #SDS-PAGE separation:

   a. Prepare the lysate sample by adding SDS reducing loading dye to ~25–30 μg of protein sample and boiling at 95°C–100°C for 5 min.

      i. Lab will record exact amount of protein loaded and provide data from determining protein concentration.

   b. Let samples cool on ice and quick-spin the tubes to collect any droplets on the cap of the tube.
   c. Prepare the gel for sample loading—insert the gel in the gel box with 1× running buffer and ensure there is no leak.

      i. Based on the expected MWs of the targets, lab will determine the optimal percentage gel to use.

   d. Load 16 μl of sample (25–30 μg/lane) in each well of the Tris–glycine gel.
   e. Run the sample at 175 V for 25 min.
   f. Remove the gel from the cassette and rinse with water.

6. #Transfer and blocking:

   a. Transfer protein on the gel to a nitrocellulose membrane for 1 hr at 12 V using a semi-dry transfer apparatus, 1× transfer buffer, and blotting sheets.
   b. Verify the efficiency of the transfer by Ponceau staining of the membrane.

      i. Lab will record an image of the Ponceau-stained membrane.

   c. Incubate the blots in 5% non-fat skim milk for 1 hr at room temperature.

7. #Antibody probing:

   a. Dilute the primary antibodies according to the manufacturer's recommendations, as suggested by the original authors.

      i. If the manufacturer recommends a range of dilutions, lab will use a dilution in the middle of the recommended dilution range.
      ii. A204:

         - p-PDGFRα.
         - PDGFRα.
         - p-AKT S473.
         - AKT.
         - p-ERK T202/Y204.
         - ERK.
         - pFRS2α Y196.
         - FRS2α.
         - β-tubulin [additional control].

            A. Loading control.

      iii. M14:

         - pHER3 Y1289.
         - HER3.

- p-AKT S473.
- AKT.
- p-ERK T202/Y204.
- ERK.
- β-tubulin [additional control].

    A. Loading control.

iv. KHM-3S:

- p-EGFR Y1068.
- EGFR.
- p-AKT S473.
- AKT.
- p-ERK T202/Y204.
- ERK.
- p-MET Y1234/5.
- MET.
- β-tubulin [additional control].

    A. Loading control.

v. HL60:

- pERK T202/Y204.
- ERK.
- β-tubulin [additional control].

    A. Loading control.

vi. MCF7:

- pHER3.
- HER3.
- β-tubulin [additional control].

    A. Loading control.

vii. HEK293:

- pMET.
- MET.
- β-tubulin [additional control].

    A. Loading control.

b. Add the antibody solutions to the membranes and incubate them for 12–16 hr at 4°C.
c. Wash the blots with Tris-buffered saline (TBS) and with 0.5% Tween-20 three times for 10 min each.
d. Dilute HRP-secondary antibody in 5% milk and add to the blots.

   i. Lab will record the dilution factor of the secondary antibody.

e. Incubate at room temperature for 1 hr.
f. Wash the blots with TBS +0.5% Tween-20 four times for 15 min each.

8. #Developing:

a. Remove as much wash buffer as possible.
b. Mix Super Signal West Pico Chemiluminescent Substrate solutions in equal proportions and add it to the blot.
c. Incubate for ~1 min.
d. Insert the blot in the developing cassette and develop the blot in the dark.

e. Expose the blot to the film at three time points, starting with 15 s. Determine the other two time points based on the strength of the signal in the 15 s exposure.

9. #Scan film and quantify band intensity using densitometric analysis software.
10. Repeat independently two additional times.

## Deliverables

- Data to be collected:

    1. Images of probed membranes (images of full films with molecular weight ladders).
    2. Scanned image of Ponceau-stained membranes after protein transfer.
    3. Quantified signal intensities and bar graphs of mean signal intensities normalized for β-tubulin loading and total pan-protein levels.

## Confirmatory analysis plan

- Statistical analysis of the Replication Data:

    1. For each cell line compare the following normalized phosphorylated kinase levels of primary kinase inhibitor alone, primary kinase inhibitor + ligand, and primary kinase inhibitor + ligand + secondary kinase inhibitor.

        - One-way ANOVA.
        - Note: at the time of analysis, we will generate a histogram of all the data to determine if it follows a Gaussian distribution or not. If it is skewed, we will perform the appropriate transformation in order to proceed with the proposed statistical analysis.

- Meta-analysis of original and replication attempt effect sizes:

    1. We will plot the replication data (mean and 95% confidence interval) and will include the original data point, calculated directly from the representative image in Figure 2C, as a single point on the same plot for comparison.

## Known differences from the original study

- We are including three additional control conditions;

    1. Media alone.

        i. To provide a baseline.

    2. Treatment of the cells with the secondary kinase inhibitor alone.

        i. To assess any effects, the secondary kinase inhibitor may be independent of the ligand and primary kinase inhibitor.

    3. Treatment of a control cell line with the growth factor ligand alone.

        i. To ensure the growth factor ligand is active.

            - FGF should cause phosphorylation of ERK1/2 in HL60 cells.
            - NRG1 should cause phosphorylation of HER3 in MCF7 cells.
            - HGF should cause phosphorylation of MET in HEK293 cells.

- The original authors recommended that the replicating lab follows a standard Western blot protocol.

## Provisions for quality control

- All data obtained from the experiment—raw data, data analysis, control data and quality control data—will be made publicly available, either in the published manuscript or as an open access dataset available on the Open Science Framework (https://osf.io/h0pnz/).

- Cell lines will be validated by STR profiling and screened for mycoplasma contamination.
- A lab from the Science Exchange network with extensive experience in conducting Western blot assays for phosphorylated proteins will perform these experiments.

## Power Calculations

### Protocol 1

The original data presented is qualitative (images of survival curves) and the authors were unable to share the raw data values with the RP:CB core team. To estimate original effect sizes, we determined approximate $IC_{50}$ concentrations from the original survival curve images.

Summary of the original data.

| A204 cells | $IC_{50}$ |
| --- | --- |
| Sunitinib | 0.05 µM |
| Sunitinib + FGF | 2.5 µM |
| Sunitinib + FGF + PD173074 | 0.025 µM |

- FGF induces resistance to Sunitinib.
- PD173074 blocks FGF-induced resistance to Sunitinib, restoring sensitivity.

| M14 | $IC_{50}$ |
| --- | --- |
| PLX4032 | 0.1 µM |
| PLX4032 + NRG1 | 0.2 µM |
| PLX4032 + NRG1 + Lapatinib | 0.1 µM |

- NRG1 induces partial resistance to PLX4032.
- Lapatinib blocks NRG1-induced resistance to PLX4032, restoring sensitivity.

| KHM-3S | $IC_{50}$ |
| --- | --- |
| Erlotinib | 0.5 µM |
| Erlotinib + HGF | >10 µM |
| Erlotinib + HGF + Crizotinib | 0.3 µM |

- HGF induces resistance to Erlotinib.
- Crizotinib blocks HGF-induced resistance to Erlotinib, restoring sensitivity.

We have calculated the projected sample size based on a variety of different possible levels of variance using a one-way ANOVA test with an alpha error of 0.05.

- These power calculations were performed with G*Power software, version 3.1.7 (*Faul et al., 2007*).
- The F statistic was calculated at http://statpages.org/anova1sm.html.
- The $\eta_P^2$ was calculated using the formula on the spreadsheet accessed from Lakens and colleagues (*Lakens, 2013*).

**A204**

| Variance | F (2, 6) | $\eta_P^2$ | Effect size f | Power | Total sample size across all groups |
| --- | --- | --- | --- | --- | --- |
| 2% | 7273.6132 | 0.999588 | 49.25631 | 99.99% | 6 |
| 15% | 129.3087 | 0.977326 | 6.565316 | 99.99% | 6 |
| 28% | 37.1103 | 0.925206 | 3.517109 | 98.53% | 6 |
| 40% | 18.184 | 0.858384 | 2.461981 | 85.32% | 6 |

For each percent variance, the relative standard deviation of the approximated $IC_{50}$ was used to calculate the F statistic from a one-way ANOVA analysis, which was converted to $\eta_P^2$ (the ratio of variance attributed to the effect and the effect plus its associate error variance from the ANOVA), and

then used to determine the effect size (Cohen's *f*) and the needed sample size to obtain at least 80% power. The actual power obtained is listed.

**M14**

| Variance | F (2, 6) | $\eta_P^2$ | Effect size *f* | Power | Total sample size across all groups |
|---|---|---|---|---|---|
| 2% | 1250 | 0.997606 | 20.4135 | 99.99% | 6 |
| 15% | 22.2222 | 0.881057 | 2.721652 | 90.90% | 6 |
| 28% | 6.3776 | 0.680089 | 1.458036 | 85.39% | 9 |
| 40% | 3.125 | 0.510204 | 1.020621 | 88.33% | 15 |

**KHM-S3**

| Variance | F (2, 6) | $\eta_P^2$ | Effect size *f* | Power | Total sample size across all groups |
|---|---|---|---|---|---|
| 2% | 6890.8212 | 0.999565 | 47.9359 | 99.99% | 6 |
| 15% | 122.5035 | 0.976096 | 6.390149 | 99.99% | 6 |
| 28% | 35.1573 | 0.921378 | 3.423315 | 98.12% | 6 |
| 40% | 17.2271 | 0.851684 | 2.396322 | 83.59% | 6 |

In order to produce quantitative replication data, we will run the experiment three times. Each time we will quantify the IC$_{50}$. We will determine the standard deviation of the IC$_{50}$ across the three biological replicates and combine this with the means from the original study to simulate an effect size. Using this simulated effect size, we will then determine the number of replicates necessary to reach a power of at least 80%. We will then perform additional replicates, if required, to ensure that the experiment has more than 80% power to detect the original effect.

## Protocol 2

The original data presented is qualitative (images of Western Blots). We used Image Studio Lite v. 4.0.21 (LICOR) to perform densitometric analysis of the presented bands to quantify the original effect size. Levels of phospho-protein were normalized to total protein and then normalized to the control.

Summary of original data.

| A204 cells | pPDGFR | pAKT | pERK | pFRS2 |
|---|---|---|---|---|
| Control | 1 | 1 | 1 | 1 |
| Sunitinib alone | 0.264 | 0.0845 | 1.952 | 1.473 |
| Sunitinib + FGF | 0.337 | 0.092 | 5.350 | 8.069 |
| Sunitinib + FGF + PD173074 | 0.304 | 0.071 | 0.369 | 1.013 |

- FGF activates pFRS2 and pERK in the presence of Sunitinib.
- PD173074 blocks FGF-induced pFRS2 and pERK activation.

| M14 cells | pHER3 | pAKT | pERK |
|---|---|---|---|
| Control | 1 | 1 | 1 |
| PLX4032 alone | 0.3667 | 1.8645 | 0.0524 |
| PLX4032 + NRG1 | 3.9447 | 11.211 | 0.0539 |
| PLX4032 + NRG1 + Lapatinib | 1.0666 | 1.7863 | 0.0571 |

- NRG1 activates pHER3 and pAKT in the presence of PLX4032.
- Lapatinib blocks NRG1-induced pHER3 and pAKT activation.

| KHM-3S cells | pEGFR | pAKT | pERK | pMET |
|---|---|---|---|---|
| Control | 1 | 1 | 1 | 1 |
| Erlotinib alone | 0.008 | 0.609 | 0.18 | 1.098 |

*Table 10. Continued on next page*

*Table 10. Continued*

| KHM-3S cells | pEGFR | pAKT | pERK | pMET |
|---|---|---|---|---|
| Erlotinib + HGF | 0.014 | 1.381 | 0.979 | 11.66 |
| Erlotinib + HGF + Crizotinib | 0.023 | 0.417 | 0.085 | 1.095 |

- HGF activates pMET and pERK in the presence of Erlotinib.
- Crizotinib blocks HGF-induced pMET and pERK activation.

We have calculated the projected sample size based on a variety of different possible levels of variance (***Koller and Wätzig, 2005***) using a one-way ANOVA test with an alpha error of 0.05.

- These power calculations were performed with G*Power software, version 3.1.7 (***Faul et al., 2007***).
- The F statistic was calculated at http://statpages.org/anova1sm.html.
- The $\eta_P^2$ was calculated using the formula on the spreadsheet accessed from Lakens and colleagues (***Lakens, 2013***).

**A204 cells**

| 2% Variance | pPDGFR | pAKT | pERK | pFRS2 |
|---|---|---|---|---|
| F(3, 8) | 2884.5133 | 6189.0064 | 4400.8341 | 5183.0738 |
| $\eta_P^2$ | 0.999076377 | 0.999569314 | 0.999394421 | 0.999485769 |
| Effect size *f* | 32.8891 | 48.17548 | 40.62403 | 44.08686 |
| Power | 99.99% | 99.99% | 99.99% | 99.99% |
| Total sample size across all groups | 8 | 8 | 8 | 8 |

| 15% variance | pPDGFR | pAKT | pERK | pFRS2 |
|---|---|---|---|---|
| F(3, 8) | 51.28023644 | 110.0267804 | 78.23705067 | 92.14353422 |
| $\eta_P^2$ | 0.950568679 | 0.976336986 | 0.967039009 | 0.971873631 |
| Effect size *f* | 4.385212 | 6.423398 | 5.416539 | 5.87825 |
| Power | 99.99% | 99.99% | 99.99% | 99.99% |
| Total sample size across all groups | 8 | 8 | 8 | 8 |

| 28% variance | pPDGFR | pAKT | pERK | pFRS2 |
|---|---|---|---|---|
| F(3, 8) | 14.71690459 | 31.57656327 | 22.4532352 | 26.44425408 |
| $\eta_P^2$ | 0.846598456 | 0.922125726 | 0.893842473 | 0.908396348 |
| Effect size *f* | 2.349221 | 3.441106 | 2.901717 | 3.149063 |
| Power | 91.97% | 99.79% | 98.43% | 99.35% |
| Total sample size | 8 | 8 | 8 | 8 |

| 40% variance | pPDGFR | pAKT | pERK | pFRS2 |
|---|---|---|---|---|
| F(3, 8) | 7.21128325 | 15.472516 | 11.00208525 | 12.9576845 |
| $\eta_P^2$ | 0.73003845 | 0.852988598 | 0.804907816 | 0.829326246 |
| Effect size *f* | 1.644455 | 2.408774 | 2.031202 | 2.204344 |
| Power | 96.95% | 93.12% | 83.18% | 88.55% |
| Total sample size across all groups | 12 | 8 | 8 | 8 |

For each percent variance, the relative standard deviation of the approximated phospho-protein level was used to calculate the F statistic from a one-way ANOVA analysis, which was converted to $\eta_P^2$ (the ratio of variance attributed to the effect and the effect plus its associated error variance from the ANOVA), and then used to determine the effect size (Cohen's *f*) and the needed sample size to obtain at least 80% power. The actual power obtained is listed.

**M14 cells**

| 2% Variance | pHER3 | pAKT | pERK |
| --- | --- | --- | --- |
| F(3, 8) | 4297.4601 | 5283.2994 | 6645.7378 |
| $\eta_p^2$ | 0.999379863 | 0.99949552 | 0.999598901 |
| Effect size $f$ | 40.14408 | 44.51111 | 49.92144 |
| Power | 99.99% | 99.99% | 99.99% |
| Total sample size across all groups | 8 | 8 | 8 |

| 15% variance | pHER3 | pAKT | pERK |
| --- | --- | --- | --- |
| F(3, 8) | 76.39929067 | 93.92532267 | 118.1464498 |
| $\eta_p^2$ | 0.966272885 | 0.972392466 | 0.977927341 |
| Effect size $f$ | 5.352545 | 5.934812 | 6.656194 |
| Power | 99.99% | 99.99% | 99.99% |
| Total sample size across all groups | 8 | 8 | 8 |

| 28% variance | pHER3 | pAKT | pERK |
| --- | --- | --- | --- |
| F(3, 8) | 21.92581684 | 26.95560918 | 33.90682551 |
| $\eta_p^2$ | 0.891565784 | 0.909977657 | 0.927087448 |
| Effect size $f$ | 2.867435 | 3.179364 | 3.565818 |
| Power | 98.24% | 99.42% | 99.88% |
| Total sample size | 8 | 8 | 8 |

| 40% variance | pHER3 | pAKT | pERK |
| --- | --- | --- | --- |
| F(3, 8) | 10.74365025 | 13.2082485 | 16.6143445 |
| $\eta_p^2$ | 0.801148125 | 0.8320201 | 0.861694667 |
| Effect size $f$ | 2.007204 | 2.225555 | 2.496073 |
| Power | 82.32% | 89.11% | 94.57% |
| Total sample size across all groups | 8 | 8 | 8 |

**KHM-S3 cells**

| 2% Variance | pEGFR | pAKT | pERK | pMET |
| --- | --- | --- | --- | --- |
| F(3, 8) | 7271.894 | 1594.1561 | 3697.7822 | 6041.5258 |
| $\eta_p^2$ | 0.999633426 | 0.998330017 | 0.999279367 | 0.999558805 |
| Effect size $f$ | 52.22032 | 24.45012 | 37.238 | 47.59802 |
| Power | 99.99% | 99.99% | 99.99% | 99.99% |
| Total sample size across all groups | 8 | 8 | 8 | 8 |

| 15% variance | pEGFR | pAKT | pERK | pMET |
| --- | --- | --- | --- | --- |
| F(3, 8) | 129.2781156 | 28.34055289 | 65.73835022 | 107.4049031 |
| $\eta_p^2$ | 0.979789525 | 0.913998523 | 0.961016505 | 0.975773338 |
| Effect size $f$ | 6.962707 | 3.260016 | 4.965066 | 6.346404 |
| Power | 99.99% | 99.57% | 99.99% | 99.99% |
| Total sample size across all groups | 8 | 8 | 8 | 8 |

| 28% variance | pEGFR | pAKT | pERK | pMET |
| --- | --- | --- | --- | --- |
| F(3, 8) | 37.1015 | 8.13344949 | 18.86623571 | 30.82411122 |
| $\eta_p^2$ | 0.932944692 | 0.753089075 | 0.876158512 | 0.920376091 |
| Effect size $f$ | 3.730022 | 1.746437 | 2.659857 | 3.399859 |
| Power | 99.94% | 98.31% | 96.62% | 99.75% |
| Total sample size across all groups | 8 | 12 | 8 | 8 |

*Table 12. Continued on next page*

*Table 12. Continued*

| 40% variance | pEGFR | pAKT | pERK | pMET |
|---|---|---|---|---|
| F(3, 8) | 18.179735 | 3.98539025 | 9.2444555 | 15.1038145 |
| $\eta_p^2$ | 0.872080242 | 0.59912149 | 0.776119611 | 0.84993841 |
| Effect size $f$ | 2.611015 | 1.222506 | 1.8619 | 2.379901 |
| Power | 96.09% | 94.83% | 99.19% | 92.58% |
| Total sample size across all groups | 8 | 16 | 12 | 8 |

In order to produce quantitative replication data, we will run the experiment three times. Each time we will quantify band intensity. We will determine the standard deviation of band intensity across the three biological replicates and combine this with the mean from the original study to simulate the original effect size. We will use this simulated effect size to determine the number of replicates necessary to reach a power of at least 80%. We will then perform additional replicates, if required, to ensure that the experiment has more than 80% power to detect the original effect.

## Acknowledgements

The Reproducibility Project: Cancer Biology core team would like to thank the original authors, in particular Dr Jeff Settleman, for generously sharing critical information to ensure the fidelity and quality of this replication attempt. We would also like to thank the following companies for generously donating reagents to the Reproducibility Project: Cancer Biology: American Type Culture Collection (ATCC), BioLegend, Charles River Laboratories, Corning Incorporated, DDC Medical, EMD Millipore, Harlan Laboratories, LI-COR Biosciences, Mirus Bio, Novus Biologicals, and Sigma-Aldrich.

## Additional information

### Group author details

**Reproducibility Project: Cancer Biology**

Elizabeth Iorns: Science Exchange, Palo Alto, United States; William Gunn: Mendeley, London, United Kingdom; Fraser Tan: Science Exchange, Palo Alto, United States; Joelle Lomax: Science Exchange, Palo Alto, United States; Timothy Errington: Center for Open Science, Charlottesville, United States

### Competing interests

EG: The Monoclonal Antibody Core Facility is a Science Exchange associated laboratory. RP:CB: EI, FT and JL are employed and holds shares in Science Exchange Inc. The other authors declare that no competing interests exist.

### Funding

| Funder | Author |
|---|---|
| Laura and John Arnold Foundation | Reproducibility Project: Cancer Biology |

The Reproducibility Project: Cancer Biology is funded by the Laura and John Arnold Foundation, provided to the Center for Open Science in collaboration with Science Exchange. The funder had no role in study design or the decision to submit the work for publication.

### Author contributions

EG, EG, Drafting or revising the article; RP:CB, Conception and design, Drafting or revising the article

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
