## [Decision Letter]

Thank you for sending your work entitled “Registered report: Widespread potential
for growth-factor-driven resistance to anticancer kinase inhibitors” for
consideration at *eLife*. Your article has been favorably evaluated by
Tony Hunter (Senior editor) and 3 reviewers, one of whom is a member of our Board of
Reviewing Editors.

The Reviewing editor and the other reviewers discussed their comments before we reached
this decision, and the Reviewing editor has assembled the following comments to help you
prepare a revised submission.

1) The experimental design to test the reproducibility of [20] is thorough and well-articulated, with some
exceptions. First, it will be important to perform positive controls to assess the
performance of the growth factors or the kinase inhibitors that will be used.

2) Second, given that the western blots in the original manuscript are not quantified
and that quantification is derived from the published work, the authors should describe
how they are going to determine whether the data are “reproducible” or
not.

3) Third, it is not immediately clear whether the distribution of the data (IC50 for
Protocol 1 and band intensity for Protocol 2) will exhibit any skew. Therefore, at the
time of analysis, it may be useful to plot histograms of the data to examine their
distributions, and, if necessary, consider suitable transformations (for example, the
Box–Cox family of transformations) of the data to obtain (approximately)
symmetric distributions so that the testing procedures are valid.

4) Lastly, the authors should either include or explain the reason for excluding in the
replication study the role of HGF-MET signaling in resistance to BRAF inhibition that
was observed in some melanomas in the original study and other reports.

---

## [Author Response]

*1) The experimental design to test the reproducibility of*
[20]
*is thorough and well-articulated, with some exceptions. First, it will be
important to perform positive controls to assess the performance of the growth
factors or the kinase inhibitors that will be used*.

We agree that verifying the activity of the reagents prior to their use in our
experiments is an important step. We have three classes of reagent: primary RTK
inhibitors, growth factor ligands, and secondary RTK inhibitors. Each cohort includes a
positive control where the cell line of interest is treated solely with its cognate
primary RTK inhibitor. This should demonstrate that the drug is active as anticipated,
and the quality control data (for both primary and secondary RTK inhibitors) provided by
the manufacturers will be included in the materials publicly available through the Open
Science Framework. However, as indicated by the reviewers, there is known lot-to-lot
variation in growth factors, so we have added steps to test the growth factors we are
using for activity. In Protocol 2, we have added in additional cell lines that have a
known response to treatment with the ligand alone, as evidenced by phosphorylation of
downstream targets. We will treat these positive control cell lines with the growth
factors and assess phosphorylation of their cognate target by Western blot. The
manuscript has been updated to reflect this additional work.

*2) Second, given that the western blots in the original manuscript are not
quantified and that quantification is derived from the published work, the authors
should describe how they are going to determine whether the data are
“reproducible” or not*.

We will present both the original data and replication data for side-by-side comparison.
We will plot the mean value of our replication data along with the 95% confidence
interval. We will then include the original data point (IC_50_ or quantified
Western blot band intensity) on the same plot to demonstrate if the original data falls
within the 95% confidence interval of the replication data. We have also updated the
language of the manuscript to reflect this change.

*3) Third, it is not immediately clear whether the distribution of the data (IC50
for Protocol 1 and band intensity for Protocol 2) will exhibit any skew. Therefore,
at the time of analysis, it may be useful to plot histograms of the data to examine
their distributions, and, if necessary, consider suitable transformations (for
example, the Box–Cox family of transformations) of the data to obtain
(approximately) symmetric distributions so that the testing procedures are
valid*.

Thank you for this suggestion. At the time of analysis, we will generate a histogram of
all the data to determine if it follows a Gaussian distribution or not. If it is skewed,
we will perform the appropriate transformation in order to proceed with the proposed
statistical analysis. We will note any changes or transformations made. We have also
updated the manuscript to address this point.

*4) Lastly, the authors should either include or explain the reason for excluding
in the replication study the role of HGF-MET signaling in resistance to BRAF
inhibition that was observed in some melanomas in the original study and other
reports*.

We agree that all of the experiments included in the original study are important, and
choosing which experiments to replicate has been one of the great challenges of this
project. In this case, the RP:CB core team felt that the most impactful information in
[20] was that bypassing RTK
inhibition by ligand-mediated activation of parallel signaling pathways was a mechanism
applicable to many different types of cancer, each with its own constellation of
addictive mutations and cognate inhibitors. The experiments addressing the role of HGF
in activating MET signaling to bypass EGFR inhibition provide a more detailed
exploration of this mechanism in one specific cancer type scenario, and support the
larger conclusion drawn from the experiments we chose for replication. As such, we will
restrict our analysis to the experiments being replicated and will not include
discussion of experiments not being replicated in this study.